# Are primary school children attending full-day school still engaged in sports clubs?

Sarah Spengler[1]*, Arvid Kuritz[2], Matthias Rabel[1], Filip Mess[1]

1 Department of Sport and Health Sciences, Technical University of Munich, Munich, Germany,
2 Department of Sports Science, University of Konstanz, Konstanz, Germany

* sarah.spengler@tum.de

## Abstract

### Purpose

Schools and organized sports both offer great chances to promote physical activity among children. Full-day schools particularly allow for extensive participation in extra-curricular physical activities. However, due to time reasons, full-day schools may also prevent children from engagement in organized sports outside school. There is only little national and international research addressing the possible competition of full-day schools and providers of organized sports outside school and the potential effects on children's physical activity behavior. In Germany's educational system, a transformation towards more full-day schools is currently taking place. The existence of both, half-day and full-day schools, gave occasion to the following research question: Do students attending half-day and full-day school differ with respect to a) sports club membership rate and b) weekly amount of sports club training?

### Methods

Data were collected in eleven German primary schools. Selected schools offered both half-day and full-day (minimum three days/week with at least seven hours) care. 372 students' data (grades 1–4; N = 153 half-day, N = 219 full-day; 47.4% male, 8.8±1.2y) were eligible for analyses. We assessed sports club membership and weekly training duration via questionnaire. Statistical analyses included Chi-square and Mann-Whitney-U-Tests.

### Results

83% of half-day school students and 67% of full-day school students were sports club members ($\chi^2(1)$ = 12.31, p<.001). Weekly duration of training in sports clubs among sports club members (N = 266) also differed between the groups (mdn = 150 min in half-day, mdn = 120 min in full-day school students; z = -2.37, p = .018). Additional analyses stratified for age and gender showed similar results.

### Conclusion

Primary school students attending full-day schools engage less in organized sports outside school than half-day school students, regardless of age and gender. Future studies should

**Data Availability Statement:** All relevant data are within the manuscript and its Supporting Information files.

**Funding:** This study was supported by funding from the Ministry of Education, Youth and Sports

Baden-Wuerttemberg, https://km-bw.de/Lde/Startseite. This work was also supported by the German Research Foundation (DFG) and the Technical University of Munich within the funding program Open Access Publishing. The funders had no role in study design, data collection and analysis, decision to publish, or preparation of the manuscript.

**Competing interests:** The authors have declared that no competing interests exist.

examine if the detected lower engagement in sports club physical activity is compensated by physical activities in other settings such as school or non-organized leisure time.

## Introduction

Already in childhood, physical activity is a key factor for health [1, 2]. Furthermore, studies show that physical activity behavior adopted in early life most often tracks into adulthood [3, 4]. Therefore, physical activity is promoted as part of a healthy lifestyle [5, 6]. Recommendations for physical activity behavior and its promotion further point to different settings in which children's physical activity should take place, i.e. in the family context, at school and in leisure time (organized sports and free play) [6].

Especially in school, physical activity programs have the potential to reach all children regardless of their socioeconomic background. However, school systems differ between countries. In most countries primary school schedules encompass four to five curriculum-based lessons but schedules differ in terms of organization: E.g. in France, Spain and England children generally spend the full day at school. In Italy parents and children can choose between a half-day and a full-day schedule [7]. In Germany until recently, primary school education mostly took place in the mornings only. However, since 2003 optional full-day school offers were introduced and since then have expanded continuously [8]. Officially accredited full-day schools have to offer at least three days a week with a minimum of seven hours of care on each day (encompassing curricular and extra-curricular activities) [9]. Numbers of students attending full-day school in Germany are continuously rising with less than 10% in 2002 and 42.5% in 2016 [9, 10]. A further expansion of full-day schools is claimed and subsidized until 2021 [11].

In primary school students, in 2016, 40.1% participated in full-day school [9]. As a result, these children spend more time at school–time in which they take part in extra-curricular programs. More than 95% of the primary schools in Germany offering full-day care include voluntary physical activity-related programs in their extra-curricular programs [12–14]. Thus, children can potentially be physically active there. They have less free time out of school though which they could spend on leisure activities such as participating in organized sports.

Next to physical education and extra-curricular physical activities in school, participating in organized sports outside school is another important and suitable way for children to be physically active [6]. Participating in organized sports increases the chances of lifelong physical activeness due to the fact that membership can survive several life changes like finishing school or changing jobs. Furthermore, performance- and competition-oriented athletes find attracting offers in organized sports. Engagement in organized sports offers a chance for social learning and integration with regard to children's development [15, 16]. Participating in organized sports is popular among children. International studies report participation rates between 36% and 66% [17–19]. However, providing institutions differ between countries. In Germany, sports clubs belong to the non-profit sector [15] and build the basis for mass sport provision [20]. Currently there are around 91,000 sports clubs which amounts to one-quarter of all third-sector organizations [21]. They are the main sports providers for the overall population in Germany. They offer affordable sports programs (monthly membership fee for children does not exceed 2.50€ in 50% of the sports clubs [22]) for competitive as well as recreational purposes [23]. German sports clubs are voluntary organizations with autonomous structures, focusing on their members' interests. They follow democratic decision

structures and rely on voluntary work [24]. The prevalence of sports club membership in children is comparatively high in Germany. In 2017, 65% of the six- to eleven-year-olds were member of a sports club [25]. Next to offering various sports programs, sports clubs pursue social tasks like educating tolerance and fair play, supporting sociability and integration [26]. Sports clubs are also seen as valuable to society due to their contribution to e.g. youth promotion, social integration, crime prevention and health [15]. Thus, again from a societal perspective, young people's participation in sports clubs is desirable. However, membership rates decrease with children's age [27, 28]. It seems especially crucial to recruit children at an early age aiming at creating a long-term commitment towards the club and by this also towards physical activity.

Both settings–school and organized sports–offer a great chance to promote physical activity in children. Longer school days in full-day schools and hence more possibilities to participate in extra-curricular physical activities can be beneficial for promoting physical activity especially in children who otherwise would not be committed to a sports club or another institution offering organized sports (e.g. children from difficult socioeconomic backgrounds [25, 29]). However, there is also the possibility that children attending full-day school will not engage in organized sports outside school due to a (perceived) lack of time or clash of dates. German sports clubs for example currently struggle to recruit members and volunteers and further fear to loose potential members and volunteers due to the expansion of full-day schools [15]. Consequently, a significantly increasing number of sports clubs started to cooperate with full-day schools (35% in 2014) [26].

From a health perspective, knowing the impact of full-day school on overall physical activity behavior in children is crucial. Furthermore, certainty about the impact of full-day school on engagement in organized sports would form the basis for developing innovative cooperation strategies that motivate children to engage in organized sports. There is only little research concerning the possible competition of full-day schools and providers of organized sports and the potential effects on children's physical activity behavior. Some studies compare physical activity levels between in-school and off-school hours (e. g. [30–34]), indicating that physical activity levels are lower in school than outside school. An Italian study [35] finds that full-day school students spend a higher percentage of time with moderate-to-vigorous physical activity (MVPA) in the afternoon timeslot than half-day school students (18% vs. 15% of MVPA). No differences are present in the morning and evening timeslots. Züchner and Arnold [36] find that sports club membership rates in German secondary school students who attend full-day school at least at two days a week are lower than in students attending half-day school only. Further, they find that in 7[th] and 9[th] graders the frequency of training in a sports club is lower in full-day school students. On the contrary, in another German study secondary school students attending full-day school offers obtain higher sports clubs membership rates than their counterparts attending half-day school only [37]. In summary, studies on the relationship of full-day school attendance and overall physical activity behavior of students are rare and results are heterogeneous. Particularly with regard to primary school students as well as to participation in organized sports there is a lack of knowledge to date.

The ongoing change in the German educational system and the existence of both half-day and full-day schools constitutes a beneficial occasion to gain more insight into this field. Due to the detected desiderata, this study aims to answer the following research question: Does habitual sports club participation differ between children attending half-day and full-day school with respect to a) sports club membership rate and b) weekly amount of sports club training?

## Methods

### Study design

The survey was conducted in eleven German primary schools in Baden-Württemberg between May and July 2017. Selected schools were officially registered and accredited as full-day school and installed an optional full-day school branch in the school year 2016/2017. Eleven out of 25 invited schools (44%) agreed to participate in the study. All students of grade 1 to grade 4 were invited to take part, regardless of their affiliation to full-day or half-day school. To collect data on students' physical activity behavior in different settings, they were asked to fill out a paper-pencil questionnaire [38] together with their parents.

Parents of each participant gave informed written consent before enrolling in the study. The study was approved by the ethics committee of the University of Konstanz, Germany and was conducted in accordance with the Declaration of Helsinki.

### Measurements

**Sex, age and attendance of half- and full-day school.**   Sex as well as date of birth was assessed via questionnaire and age was calculated. Two age groups were built by using the median age (8.79 years) as cut off for dividing the sample.

Within the questionnaire participants were asked if they were registered for full-day school (i.e. these children spend at least seven hours a day at three days a week in school, according to the accreditation requirements). If they were not registered, they were further asked if they nevertheless attended full-day school offers. For both groups (registered and non-registered), time spent at full-day school offers was assessed for each day during the week of the assessment. Participants wrote down the exact time when they left school at each day they attended full-day school offers. Participants were divided into two groups regarding their program: Full-day and half-day group. All children that were registered for full-day school were allocated to the full-day group. Children who answered that they were not registered and neither attended full-day offers were allocated to the half-day group. Children who were not registered for full-day school but attended full-day offers which cover–in combination with regular lessons–at least three days with a minimum of seven hours at school were also allocated to the full-day group. This cut-off-point is in accord with the requirements to be accredited and financially supported as full-day school in Germany. Children who attended lower amounts of full-day school offers were not included in the analyses.

**Habitual sports club participation.**   Habitual sports club participation was assessed using the scale from the MoMo Physical activity questionnaire (wave 2) [38]. Sports club participation in Germany standardly includes regular training sessions, commonly on a weekly basis throughout the whole year. Performing sports in a sports club can therefore be seen as a habitual physical activity with fixed weekly practices that take place throughout the whole year. Exceptions can be seasonal sports, e.g. skiing. Participants were asked if they are member of at least one sports club. Further, they were asked how much and which physical activity they exert in sports clubs. The questionnaire included questions considering type (which sport) of their physical activity, weekly duration (in minutes) of each performed sport and months in which each sport was performed throughout the year (to detect seasonal activities). Participants could report data for maximal four different types of sport in sports clubs. From these data, a sum score was calculated reflecting the duration of performed physical activity in sports clubs per week. The scale assessing sports club participation was checked in relation to reliability (kappa = .81) and validity (correlations between scale and Actigraph GT1M: r = 0.35) [39]. However, this validation is based on data of adolescents aged 11 to 17 years and on a former

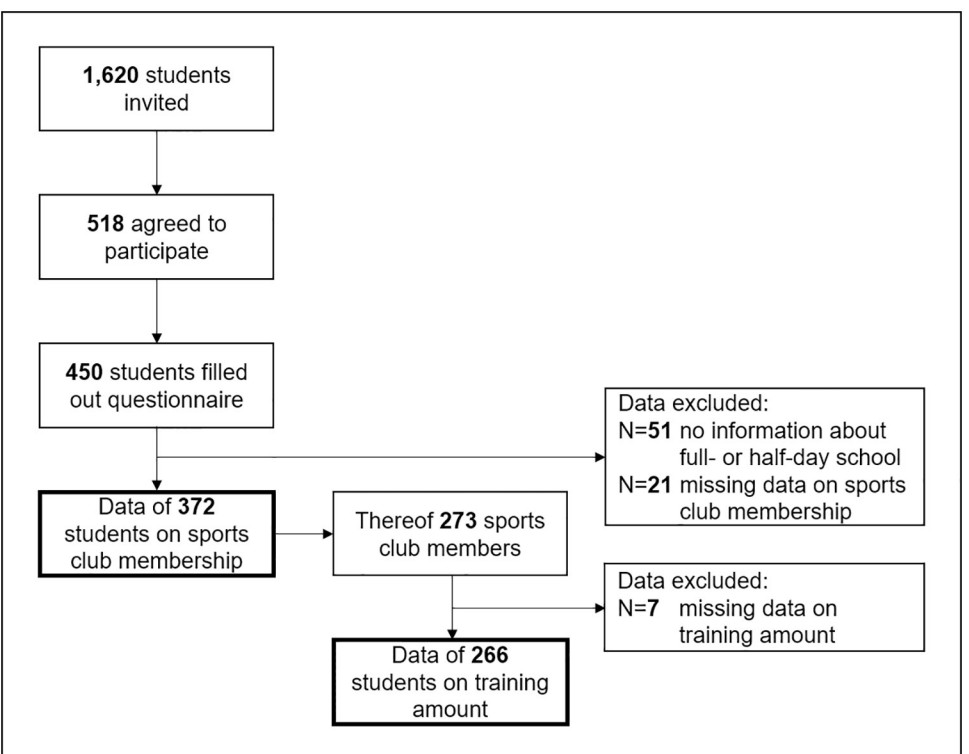

**Fig 1. Flow chart of the sample selection process.** Bold frames indicate final sample sizes for the statistical analyses.

version of the scale. The scale used here reflects an optimized version (with slight modifications) published by the authors of the MoMo Study [38].

## Participants

Out of 1,620 invited students, 518 (32.0%) agreed to participate in the study. 450 participants filled out the questionnaire (46.8% male, mean age 8.8±1.2y). Fig 1 shows how the final sample size was achieved. Table 1 displays the final sample sizes for the analyses of each part of the research question and shows the students' distribution in grades 1 to 4.

## Statistics

All statistical tests were performed in SPSS statistical software for Windows (release 23.0; SPSS Inc., Chicago, IL, USA). Pearson's chi-square test of independence was used to identify differences in sports club membership rate between groups. Mann-Whitney-U tests were performed to compare weekly duration of training in sports clubs between groups. The significance level for all statistical tests was set a priori to $\alpha \leq .05$.

## Results

Membership rates in sports clubs differed significantly between students attending half-day and full-day school (Fig 2). 83% of the students attending half-day school were member in a sports club compared to 67% of the students attending full-day school ($\chi^2(1) = 12.31$, p<.001). Additional analyses stratified for students' gender and age showed similar patterns for boys and girls as well as for younger and older students: Regarding boys, 90.8% (half-day) and

**Table 1. Sample sizes.**

|  | Sports club membership (N) | Habitual PA in sports clubs (N) |
|---|---|---|
| **Grade 1** | 45 (m) / 35 (f) | 33 (m) / 26 (f) |
| **Grade 2** | 29 (m) / 55 (f) | 23 (m) / 35 (f) |
| **Grade 3** | 53 (m) / 42 (f) | 44 (m) / 30 (f) |
| **Grade 4** | 27 (m) / 40 (f) | 18 (m) / 23 (f) |
| **Inter-grade or no information about grade**[a] | 23 (m) / 23 (f) | 17 (m) / 17 (f) |
| **Average age** | 8.8±1.2 (m) / 8.7±1.2 (f) | 8.8 ±1.1 (m) / 8.6±1.1 (f) |
| **Total** | 372 | 266 |

[a] These students attended an inter-grade class (different combinations: grades 1 and 3; 2 and 4; 1 and 2; 3 and 4); 3 students (100% male) did not mention their grade in the questionnaire.

71.4% (full-day) were member of a sports club ($\chi^2(1) = 9.13$, p = .003) Regarding girls, 77.3% (half-day) and 61.7% (full-day) were member of a sports club ($\chi^2(1) = 5.46$, p = .019). Further, 85.1% of younger students attending half-day school and 66.4% of younger students attending full-day school were sports club members ($\chi^2(1) = 8.03$, p = .005). In older students, 80.8% (half-day) and 67.0% (full-day) were member of a sports club ($\chi^2(1) = 4.17$, p = .041).

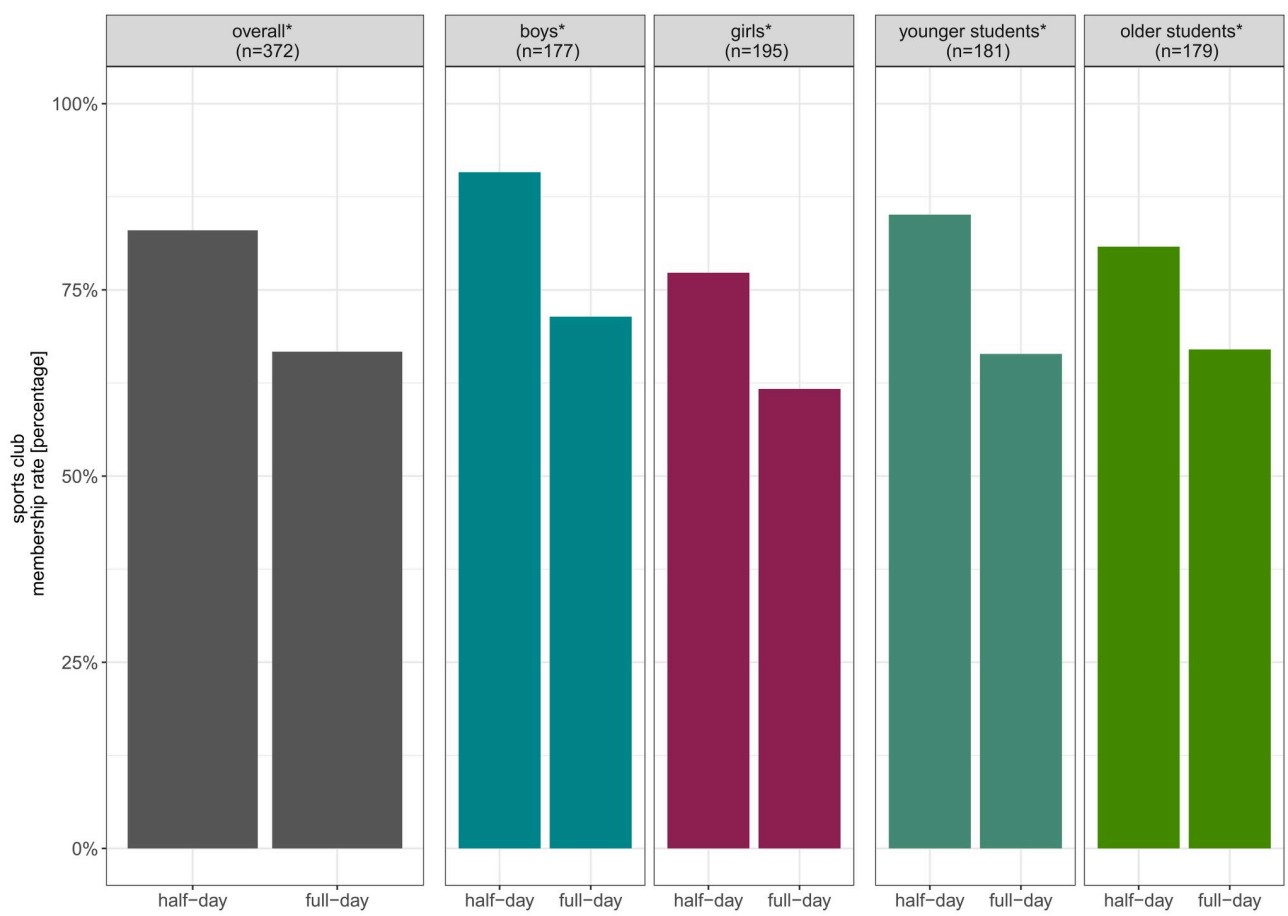

**Fig 2. Sports club membership rates overall and stratified for gender and age group (N = 372).** * indicates a statistically significant difference on the alpha-level $\alpha \leq .05$.

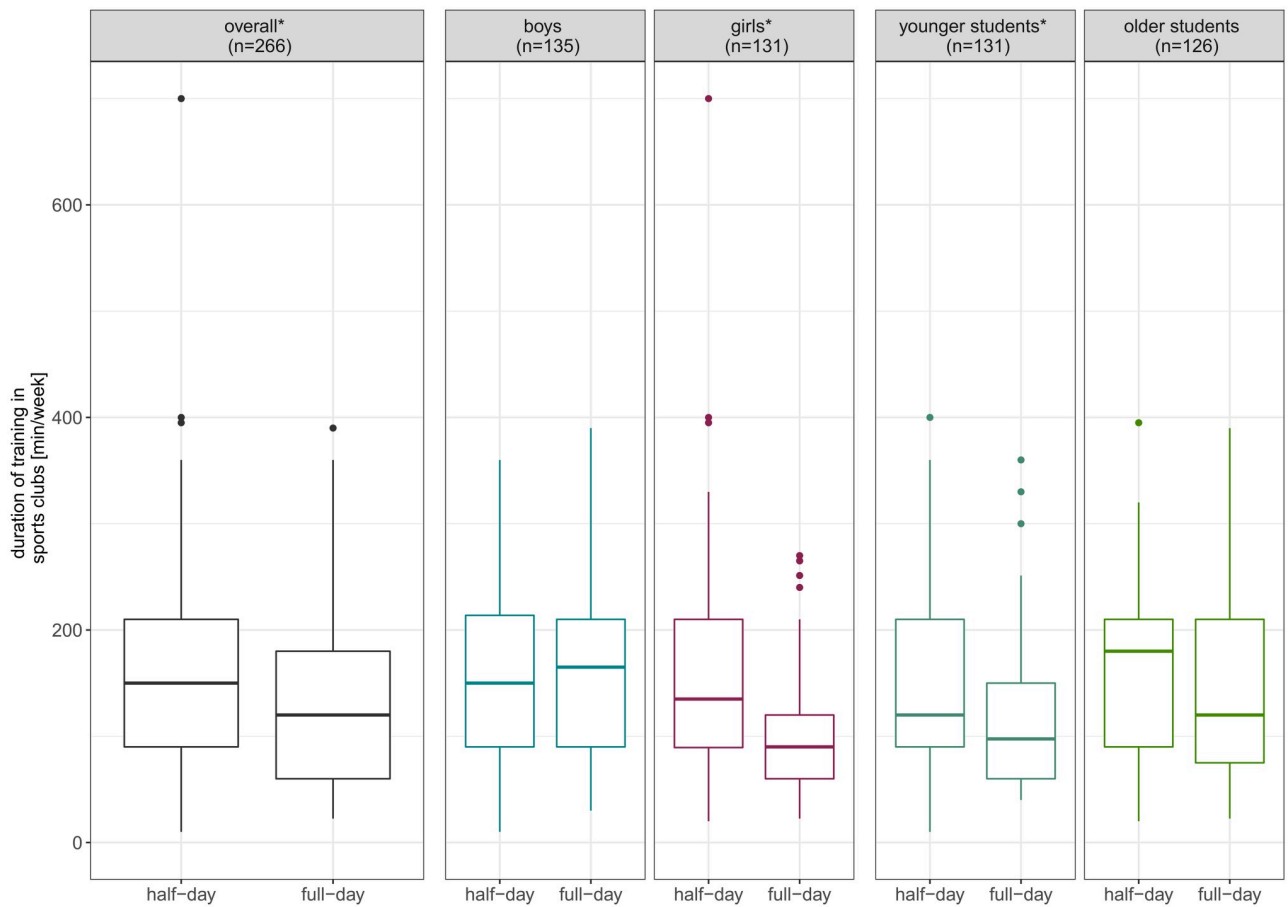

**Fig 3. Duration of training in sports clubs per week (Mdn) overall and stratified for gender and age group (N = 266).** The boxplots display the median and the 2$^{nd}$ and 3$^{rd}$ quartile. * indicates a statistically significant difference on the alpha-level $\alpha \leq .05$.

Including only sports club members (N = 266, Fig 3), our analyses showed a significant difference in weekly duration of training in sports clubs with a median of 150 min for students attending half-day school and a median of 120 min for students attending full-day school (z = -2.37, p = .018, r = .15). Stratified additional analyses for boys revealed no significant difference with medians of 150 min (half-day) and 165 min (full-day, z = -.28, p = .821, r = .02). Duration of training among girls differed significantly with medians of 135 min (half-day) and 90 min (full-day, z = -3.51, p<.001, r = .31). In the younger age group, duration of training was also significantly higher for half-day school students (mdn = 120 min) than for full-day school students (mdn = 97.5 min, z = -2.11, p = .035, r = .18). In the older age group, differences were not significant but medians were 180 min for half-day school students and 120 min for full-day school students (z = -1.15, p = .249, r = 0,10).

## Discussion

This study aimed to identify potential differences in participation in organized sports (i.e. sports clubs) between primary school students attending half-day or full-day school. We found that full-day school students were less engaged in sports clubs. This holds true for boys and girls as well as younger and older students.

Our result that boys were more often member of a sports club than girls, is in line with other studies in this field [15, 27, 40]. Our results further confirm findings that membership rates decline with children's age [27, 28], as the prevalence was already lower in the older age group. Interestingly, overall membership rates in our study were higher than reported in other national studies [25, 27]. One possible explanation could be the fact that the participating schools in our study are located in well situated areas of Southern Germany (Baden-Württemberg) with a relatively high mean net equivalent income [41]. The proportion of people with a medium or high socioeconomic status (SES) therefore might be relatively high in our sample. It is shown that children from difficult socioeconomic backgrounds tend to be less engaged in organized sports [25, 29, 42, 43]. These children were potentially underrepresented in our sample.

Our results indicate that participating in full-day school prevents some children from engaging in organized sports outside school. One reason could be that these children already attend extra-curricular physical activity programs at school and therefore do not want to engage in further organized physical activity offers. Another reason might be a lack of time due to the longer school day. It can be assumed that these children or their parents are not willing to realize further binding activities during the week in addition to the fixed full-day schedules. There is also the possibility that a clash of dates prevents these children from engaging in a sports club [37], as especially in young children sports club training often takes place in the afternoon. Similar results are found in secondary school students [36], with full-day school students obtaining lower membership rates in sports clubs though with smaller differences between the groups. One reason might be that sports clubs offers for older children and adolescents usually take place in the evening and therefore do not overlap with full-day school schedules. Furthermore, students' age could be a relevant factor with younger children being more protected by their parents from having huge amounts of bonded time, especially with regard to the fact that younger children are more likely to engage in free play [25]. This assumption holds also true for our results with a slightly greater difference in sports club membership prevalence between full-day and half-day school students in the younger age group. Heim and colleagues' [37] contradictory results might be due to the fact that in their study students were allocated to the full-day group even if they attended full-day school only one day per week, which in consequence might not affect their engagement in sports clubs.

Even if full-day school students decide to participate in sports club physical activity they spend a lower amount of weekly time exercising. Possible reasons might be similar to the reasons mentioned before: Attended physical activity programs already at school, a clash of dates and the desire not to have too many fixed appointments during the week. Again, Züchner and Arnoldt [36] show similar results for 7th and 9th graders, with full-day school students participating in less weekly training sessions than half-day school students. However, we found such results in girls but not in boys and assume that if boys decide to participate in organized sports, the performance motivation is more prevalent than in girls and hence they participate more frequently. Studies on motivation towards sports support this assumption, as it is shown that for boys performance is an important motivator for engaging in sports and physical exercise [44]. The fact that the weekly amount of training in our study was lower in girls and in addition the variance was quite low compared to boys, indicates that in female full-day school students the pattern of engaging in sports clubs in a manageable timeframe is typical. A similar picture was found in younger students. It seems as if especially young kids as well as girls might be less willing and/or be more protected by their parents from having several fixed appointments and liabilities. A recent study from 2017 supports this assumption with regard to age differences showing that younger children (six to seven years) participate less also in

other (not physical activity-related) clubs and extra-curricular groups than eight to eleven year-olds (68% vs. 80%), but engage more often in free play at home [25].

In summary, students attending full-day school engage less in sports clubs both with regard to membership rate and weekly duration of training. Sports clubs quite often cooperate with schools with the aim of recruiting members [13] but this does not seem to fully compensate the potential loss of members due to full-day school. However, type and amount of cooperation between sports clubs and schools are diverse [45]. Cooperation could include constant engagement in the full-day extra-curricular program or only one-time events to inform students about the offers of the respective club. Studies show that between 30% and 80% (depending on the studied region and schools) of the extra-curricular physical activity programs in full-day school are organized and held by sports clubs [13, 45]. However, only a small number of sports clubs (between 5% and 24%, depending on the studied region) cooperates with full-day schools in terms of extra-curricular physical activity programs [45]. Possibly the participation in physical activity programs at school (organized by other providers than sports clubs) might prevent students from engaging in sports clubs–and maybe also from engaging in leisure time physical activity in general. This assumption is supported by a study on sports schools (with compulsory physical education on each school day) which shows that primary school students attending sports schools are more active at school but less active in leisure time than students from regular schools, resulting in similar overall activity levels [46]. Hence, from a sports club perspective, it could be beneficial to expand the cooperation between sports clubs and full-day schools, both in terms of quantity and intensity.

From a health perspective, it is possible that children compensate the lower amount of physical activity, which might result from fewer engagement in sports clubs [47]. A potential compensation can be realized in several settings. One could be full-day school itself. Participation in extra-curricular programs including physical activity could compensate the lower engagement in sports club. In German schools, more than half of the full-day school students participate in extra-curricular physical activity programs [37, 48]. Pau et al.'s study supports this idea showing that in the afternoon timeslot percentage of MVPA is higher in full-day school students than in half-day school students [35]. The authors further hypothesize that this might result from a one-hour recess in the afternoon in which primary school students accumulate MVPA. Other studies also show that recess time is an important contributor to MVPA [49, 50]. Another possibility to compensate lower engagement in organized sports is being physically active in free leisure time (e.g. active outdoor play), whereby Pau et al. don't find differences in leisure time physical activity levels between half-day and full-day school students, however [35]. Both, being physically active in recess as well as in leisure time, though happens unorganized and unregularly. Therefore physical activity levels might be lower than in organized sports clubs programs, which are held by an educated trainer and take place regularly at least once a week (independent from e.g. weather and other external as well as internal barriers). Due to a lack of studies on this topic, these considerations must be examined by future scientific studies analyzing not only the quantity of children's physical activity behavior, but also the respective settings where physical activity is accumulated.

This is the first study addressing the potential competition of full-day schools and organized sports considering primary school children. However, some limitations must be considered when interpreting the findings. First, this study was based on regional data which might not be representative–neither for Germany nor for the state of Baden-Württemberg–due to regional differences, e.g. SES of the inhabitants, school size as well as a particular structure of sports clubs in the different communities and towns where the schools are located [25, 41]. Second, we did not assess SES of the students and therefore cannot fully preclude that SES was a confounder in our study. Indeed, sports club membership is significantly lower in children and

adolescents with a lower SES– 42.8% (low), 61.0% (medium) and 74.1% (high) [43]. However, a representative study on full-day school attendance in (western) German primary schools shows that the distribution of children from low, medium and high SES does not significantly differ between full-day and half-day schools [51]. Thus, we can assume that our full-day and half-day samples are similar regarding their SES distribution. Third, as data was assessed via questionnaire, statements on physical activity behavior can be affected by the difficulty to recall the duration of activities and summarizing as well as rounding this information. Questionnaire data was used because it delivers information not only with respect to the quantity of physical activity but also with respect to the setting in which the activity takes place.

## Conclusions

This study showed that primary school students attending full-day school engage less in organized sports outside school than half-day school students. Future studies should examine if the lower engagement in physical activity in sports clubs is compensated in other settings like school or leisure time outside sports clubs. Sports clubs could intensify the cooperation with schools in order to recruit long-term members and thus minimize the loss of members due to full-day school. This is especially important as engagement in sports clubs can continue even after graduation and thus contributes to continuous physical activity in adolescents and young adults.

## Supporting information

**S1 File. Study dataset.**
(XLSX)

## Acknowledgments

The authors thank Melina Schnitzius (Technical University of Munich, Germany) for copy editing the manuscript.

## Author Contributions

**Conceptualization:** Sarah Spengler, Arvid Kuritz, Filip Mess.

**Data curation:** Arvid Kuritz, Matthias Rabel.

**Formal analysis:** Sarah Spengler, Matthias Rabel.

**Funding acquisition:** Filip Mess.

**Investigation:** Arvid Kuritz.

**Methodology:** Sarah Spengler, Matthias Rabel.

**Project administration:** Arvid Kuritz, Filip Mess.

**Supervision:** Filip Mess.

**Validation:** Sarah Spengler.

**Visualization:** Sarah Spengler, Matthias Rabel.

**Writing – original draft:** Sarah Spengler.

**Writing – review & editing:** Sarah Spengler, Arvid Kuritz, Matthias Rabel, Filip Mess.

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
