## [Decision Letter · Decision Letter 0]

30 Aug 2019

PONE-D-19-19689

Are primary school children attending full-day school still engaged in sports clubs?

PLOS ONE

Dear Dr Spengler,

Thank you for submitting your manuscript to PLOS ONE. After careful consideration, we feel that it has merit but does not fully meet PLOS ONE’s publication criteria as it currently stands. Therefore, we invite you to submit a revised version of the manuscript that addresses the points raised during the review process.

We would appreciate receiving your revised manuscript by Oct 14 2019 11:59PM. To enhance the reproducibility of your results, we recommend that if applicable you deposit your laboratory protocols in protocols.io, where a protocol can be assigned its own identifier (DOI) such that it can be cited independently in the future. For instructions see: http://journals.plos.org/plosone/s/submission-guidelines#loc-laboratory-protocols

We look forward to receiving your revised manuscript.

Kind regards,

David Meyre

Academic Editor

PLOS ONE

Journal Requirements:

2. Please update your ethics statement on the online submission information to include the details of parental consent given in the manuscript.

Reviewers' comments:

Reviewer's Responses to Questions

**Comments to the Author**

1. Is the manuscript technically sound, and do the data support the conclusions?

Reviewer #1: Partly

Reviewer #2: No

2. Has the statistical analysis been performed appropriately and rigorously? 

Reviewer #1: Yes

Reviewer #2: No

3. Have the authors made all data underlying the findings in their manuscript fully available?

Reviewer #1: Yes

Reviewer #2: Yes

4. Is the manuscript presented in an intelligible fashion and written in standard English?

Reviewer #1: No

Reviewer #2: No

5. Review Comments to the Author

Reviewer #1: Review of Manuscript: Are primary school children attending full-day school still engaged in sports clubs?

This paper represents an account of how many primary-school age students participate in organized sports in sports clubs in Germany, and whether this differs based on their enrollment in full-day or half-day school programs. Furthermore, it also investigates whether those that are members of sports clubs engage more or less depending on their allocation to full- or half-day school.

While this paper is informative of the simple research question it set out to answer, I am not sure it is rigorous enough to warrant publication at this time. Major issues here are the inability to account for confounding factors, that they identify and discuss in the discussion. For example, they did not measure leisure time PA or PA/sport participation in school. They did not record or account for social-economic status and how this may affect sports club participation (presumably it costs money to belong to a sports club – this is a big confounder). Although they do state that this is a more affluent area of Germany, it is still important missing data. These omissions represent large flaws in the data analysis, and I am not sure how the researchers can address them if they did not collect these data (except for discussing their theoretical influence - which they have. I am just not sure this is enough).

Other areas to consider fixing:

1) The introduction is too long (4.5 pages). Should be cut in half and focused on the most important information.

2) First sentence of 2nd paragraph… “physical activity has the potential…” (or offers…)

3) The timeframe of data collection is very short. Or maybe I am misunderstanding… perhaps more details on the questions they were asked in the questionnaire is warranted. What period of time were they asked to evaluate? Sports club participation just from May to July, or over the year? In the last 6 months? Please add more details here. Perhaps the seasons/weather etc. makes a difference? I see that this was mentioned, but more details here would be helpful.

4) Is 11 schools representative?

5) Are the sports clubs in close proximity to the schools? Easy to get to? Are there other structural barriers?

6) Participants: should be 1620 or 1,620 not 1.620 (and elsewhere in the paper).

7) Statistics, should be α≤.05

8) Consider reporting statistics on the figures (as stars, etc.) and in the figure legends so that the figures stand alone.

9) Figure 2, I think the y-axis should be min/week.

10) Would be beneficial to better describe the families demographically in terms of the groups. i.e. stats between the full vs. half day school families? (income, # of children, etc.).

11) Would be a good idea, early on, do thoroughly define what a “sports club” in Germany is? International readers may be thinking different things. Is it like a community centre? A private club? A country club? Etc.

12) Several typographical and grammatical errors. Would benefit from additional proofreading.

Reviewer #2: The study by Sarah Spengler explored whether primary school children attending full-day schools still engaged in sports clubs. The topic is interesting and seems important in Germany, as number of children attending full-day schools has increased with high speed. However, there are some consideration as follows;

(1) The conception of “sports club” is ambiguous. Does it mean all kinds of sports club or out-school sports club? It seems that only out school sports club is the object, but it needs to be clarified in study purpose.

(2) Why is out-school sports club so important for children’s early development? Besides a few sentences in introduction, the lack of statistical analysis and in-depth discussion made the argument unconvinced to readers.

(3) There were too more unnecessary descriptions in Introduction and Discussion, but not enough content of results. I suggest the authors 1) reword the Introduction carefully, conclude the past studies with highly condensed sentences and directly bring up the importance of the present study. 2) make more statistical analysis to expand the present results. 3) try to avoid duplications of results in Discussion, state the principle finding of the study only, make more in-depth discussion about how the present study relates to past studies and how the present study adds new views to the thesis.

For example, the great length is used to indicate those missing numbers of each LS7 component from line 1 to line 17 on page 9. These should be detailed using a flow chat.

(4) The symbol within numbers should be “,” instead of “.”.

6. PLOS authors have the option to publish the peer review history of their article (what does this mean?). If published, this will include your full peer review and any attached files.

Reviewer #1: No

Reviewer #2: No

---

## [Author Response · Author response to Decision Letter 0]

14 Oct 2019

We would like to thank the reviewers for their helpful comments and the opportunity to improve our manuscript. We believe that the quality of our manuscript has substantially improved by implementing the suggestions and comments received. Please find our response to the reviewers’ comments below. The reviewers’ comments are shown in quotation marks, and our responses are shown below each comment. We hope we have addressed all the reviewers’ concerns and comments to their satisfaction.

Review Comments to the Author

Reviewer #1

“This paper represents an account of how many primary-school age students participate in organized sports in sports clubs in Germany, and whether this differs based on their enrollment in full-day or half-day school programs. Furthermore, it also investigates whether those that are members of sports clubs engage more or less depending on their allocation to full- or half-day school.

While this paper is informative of the simple research question it set out to answer, I am not sure it is rigorous enough to warrant publication at this time. Major issues here are the inability to account for confounding factors, that they identify and discuss in the discussion. For example, they did not measure leisure time PA or PA/sport participation in school. They did not record or account for social-economic status and how this may affect sports club participation (presumably it costs money to belong to a sports club – this is a big confounder). Although they do state that this is a more affluent area of Germany, it is still important missing data. These omissions represent large flaws in the data analysis, and I am not sure how the researchers can address them if they did not collect these data (except for discussing their theoretical influence - which they have. I am just not sure this is enough).”

Thank you for highlighting the paper’s limitations – we totally understand your comment. We acknowledge the fact that it would be highly interesting to find out if the full-day students compensate their lower sports clubs engagement in other PA settings. However, sports clubs are by far the main providers of physical activity offers in Germany (also see our revision regarding information on function and value of sports clubs in Germany in the introduction). Hence, we consider the findings of this paper as relevant and noteworthy on their own as they directly affect sports clubs’ recruitment of new members and sports clubs’ success. Nevertheless, further research should aim to answer the question if full-day school students’ lower engagement in sports clubs might be compensated by physical activity in other settings. This is especially important from a health perspective. 

Further, we want to address the mentioned concerns regarding socio-economic status (SES) and how it may affect sports club membership: Indeed, sports club membership is significantly lower in children and adolescents with a lower SES – 42.8% (low), 61.0% (medium) and 74.1% (high) . However, a representative study examining full-day school participation in German primary schools shows that the distribution of children from a low, medium and high SES does not significantly differ between full-day and half-day schools. Thus, we can assume that our full-day and half-day samples are similar regarding their SES distribution. Therefore SES does not seem to be a confounder with respect to our research question. We hope that we explained these circumstances to your satisfaction and included this valuable information in our manuscript. 

“Other areas to consider fixing:

1) The introduction is too long (4.5 pages). Should be cut in half and focused on the most important information.”

Thank you very much for this comment. We shortened the introduction focusing on the most important information. 

“2) First sentence of 2nd paragraph… “physical activity has the potential…” (or offers…)”

Thank you for this comment. We agree that the sentence includes misleading wording and reworded it. 

“3) The timeframe of data collection is very short. Or maybe I am misunderstanding… perhaps more details on the questions they were asked in the questionnaire is warranted. What period of time were they asked to evaluate? Sports club participation just from May to July, or over the year? In the last 6 months? Please add more details here. Perhaps the seasons/weather etc. makes a difference? I see that this was mentioned, but more details here would be helpful.”

Thank you for this comment. We revised the part on data collection timeframe as well as on how we assessed habitual sports club participation to resolve this issue. Furthermore, also referring to one of your later comments, we added information on the tradition of sports clubs in Germany in the introduction, which may also help to understand the issue. 

Sports club participation in Germany standardly includes regular training sessions, commonly on a weekly basis throughout the whole year. Performing sports in a sports club can therefore be seen as habitual physical activity with fixed weekly practices that takes place throughout the whole year. Exceptions can be seasonal sports, e.g. skiing.

Therefore the questionnaire gathers habitual sports club participation by assessing the duration of training sessions per week. To detect seasonal activities the students were also asked to sign the months (Jan to Dec) in which the training sessions take place. Hence, the questions refer to the whole year. 

Data collection took place between May and July 2017. Within this timeframe we visited the schools to conduct the survey. 

“4) Is 11 schools representative?”

Thank you for this comment. As mentioned in the limitations section, our data is not representative. The study was based on regional data, which might not be representative due to regional differences, e.g. regarding SES of the inhabitants as well as a diverse range of sport activities in sports clubs. We added more information to further clarify the issue in the limitations section of the manuscript’s revised version.

“5) Are the sports clubs in close proximity to the schools? Easy to get to? Are there other structural barriers?”

Thank you for this remark. We agree that knowing these facts is important to better understand the paper: In Germany, there are currently around 91,000 sports clubs (and 15,000 primary schools) with many of them offering different types of sports to their members. Almost every small village has at least one sports club and children’s membership fees usually do not exceed 3€ per month. Thus, structural as well as financial barriers are rather small.

“6) Participants: should be 1620 or 1,620 not 1.620 (and elsewhere in the paper).”

Please excuse this mistake. We resolved it. 

“7) Statistics, should be α≤.05”

Please excuse this mistake. We resolved it. 

“8) Consider reporting statistics on the figures (as stars, etc.) and in the figure legends so that the figures stand alone.”

Thank you for this suggestion, which we happily implemented. 

“9) Figure 2, I think the y-axis should be min/week.”

Thank you for this remark. We revised it as suggested. 

“10) Would be beneficial to better describe the families demographically in terms of the groups. i.e. stats between the full vs. half day school families? (income, # of children, etc.).”

Thank you for this suggestion. Unfortunately, we did not have the permission to assess data on the families’ demographic data. However, as mentioned in our answer to your first comment, a representative study examining full-day school participation in German primary schools shows that the distribution of children from low, medium and high SES does not significantly differ between full-day and half-day schools, neither does the number of children. Thus, we can assume that our full-day and half-day samples are similar regarding their SES distribution. We also included this information in our paper. 

“11) Would be a good idea, early on, do thoroughly define what a “sports club” in Germany is? International readers may be thinking different things. Is it like a community centre? A private club? A country club? Etc.”

Thank you for this important suggestion. We agree that a comprehensive understanding on the tradition and function of sports clubs in Germany is necessary to interpret the study’s findings as well as to understand the relevance of the paper. Therefore, we included a description of German sports clubs in the introduction. 

“12) Several typographical and grammatical errors. Would benefit from additional proofreading.”

Thank you for your feedback. We edited the manuscript.

Reviewer #2: 

“The study by Sarah Spengler explored whether primary school children attending full-day schools still engaged in sports clubs. The topic is interesting and seems important in Germany, as number of children attending full-day schools has increased with high speed. However, there are some consideration as follows;

(1) The conception of “sports club” is ambiguous. Does it mean all kinds of sports club or out-school sports club? It seems that only out school sports club is the object, but it needs to be clarified in study purpose.”

Thank you very much for this comment. Sports clubs in Germany describe organizations outside school. We agree that a comprehensive understanding on the tradition and function of sports clubs in Germany is necessary to interpret the study’s findings as well as to understand the relevance of the paper. Therefore, we included a description of the conception of German sports clubs in the introduction.

“(2) Why is out-school sports club so important for children’s early development? Besides a few sentences in introduction, the lack of statistical analysis and in-depth discussion made the argument unconvinced to readers.”

Thank you for this important remark. We added a more in-depth explanation of sports clubs in Germany, including their tradition and their relevance, in the introduction. We hope that this strengthens our argumentation. 

“(3) There were too more unnecessary descriptions in Introduction and Discussion, but not enough content of results. I suggest the authors 1) reword the Introduction carefully, conclude the past studies with highly condensed sentences and directly bring up the importance of the present study. 2) make more statistical analysis to expand the present results. 3) try to avoid duplications of results in Discussion, state the principle finding of the study only, make more in-depth discussion about how the present study relates to past studies and how the present study adds new views to the thesis. For example, the great length is used to indicate those missing numbers of each LS7 component from line 1 to line 17 on page 9. These should be detailed using a flow chat.”

Thank you very much for these remarks. As proposed, we condensed the introduction. Further, we revised the discussion avoiding duplications and focused on relations to past studies and our study’s implications. 

Unfortunately, we could not identify matching content to the remark “the great length is used to indicate those missing numbers of each LS7 component from line 1 to line 17 on page 9” at the given page. Maybe the reviewer’s version of the manuscript had different page and line numbers than ours. We suppose that the comment refers to the explanation in the methods section on how we reached the final sample size. We therefore changed this part as suggested by including a flow chart and shortening the text.

We also want to reply to the suggestion to “make more statistical analysis to expand the results”. We analyzed if half-day and full-day school students differ in terms of sports club membership and training duration and stratified the analyses for age and gender. We agree that it would be of interest to analyze further variables next to age and gender regarding their potential confounding effects. Unfortunately, as we pointed out in the limitations paragraph, we do not have the data to conduct these analyses. We addressed this problem by referring to a study showing that SES had no effect on full-day or half-day school membership. Nevertheless, we believe that we were able to answer our research question with the conducted statistical analyses. Furthermore, the multiplication of statistical models generates alpha risk inflation. Our study is one of the first to analyze the impact of full-day school on sports club participation, using well pheno-typed data and therefore adds important knowledge on this topic. In our view, this study can be seen as a first explorative step towards understanding the impact of a long(er) school day on children’s physical activity behavior. 

“(4) The symbol within numbers should be “,” instead of “.”.”

Please excuse this mistake. We resolved it. 

Reviewer #null:

“1) Introduction. Comprehensive “

Thank you. 

“2) Methods

a) Sample. Not respresentative of the schoolds in the city (44% schools accepted). Please comment.”

As already stated in the limitations section, we agree that our data is not representative. We added another explanatory sentence in the limitations section to further clarify the issue. 

“b) Use of MoMo questionnaire to assess sport’s club participatio:

 This questionnaire was validated at older ages, than the ones studied in the manucript.

 Authors shoud comment on this aspect, as studying younger ages may compromise the quality of the report.

On the other hand, the validity found fro this questionnaire is low (0.33)”

We totally agree that the validity of the questionnaire is low (correlations between scale and Actigraph GT1M: r=0.35). However, these results were similar to those of other questionnaires measuring physical activity in children and adolescents . Statements in questionnaires on physical activity can be affected by the difficulty to recall the duration of activities and to summarize as well as round this information. Choosing questionnaires to collect data was predetermined by our aim to get information on habitual physical activity exclusively in the sports club setting which would not have been measureable via accelerometer. 

Further, we acknowledge the fact that the questionnaire was validated with older participants and thus our results must be interpreted with care. Unfortunately, to the best of our knowledge, there does not exist another tool, which is able to measure habitual sports clubs participation in such detail. The MoMo questionnaire is a widely used and accepted tool in German literature and is also used in nation-wide longitudinal representative studies on physical activity behavior of primary school-aged children , .

“3) Results

a) Please indicate what d you refer as “younger ages”

Thank you for this comment. As stated in the methods section, two age groups were built by using the median age as cut off for dividing the sample. Thus, the younger age group includes children younger than 8.79 years. We included this information in the manuscript. 

“b) No indication of the time of physical activity within the school (number of hours of Physical Education classes), which could be diffrent given socio-economic status.

c) Nor indication either of time in games in recess or free time.”

Thank you for this feedback. We agree that the lower engagement in sports clubs of full-day school students might be compensated by being physically active in other settings, i.e. in voluntary physical activity programs at full-day school, in recess or in free leisure time out of school. However, sports clubs are by far the main providers of physical activity offers in Germany. Hence, we consider the findings of this paper as relevant and noteworthy on their own. Nevertheless, further research should aim to answer the question if full-day school students’ lower engagement in sports clubs might be compensated by physical activity in other settings.

“Other aspects

Phrases should not be initiated with a preposition (for, then, or …etc).”

Thank you for this remark. We carefully checked the manuscript and avoided initiating phrases with a preposition.

---

## [Editor Report · Decision Letter 1]

31 Oct 2019

Are primary school children attending full-day school still engaged in sports clubs?

PONE-D-19-19689R1

Dear Dr. Spengler,

We are pleased to inform you that your manuscript has been judged scientifically suitable for publication and will be formally accepted for publication once it complies with all outstanding technical requirements.

With kind regards,

David Meyre

Academic Editor

PLOS ONE
---

## [Editor Report · Acceptance letter]

11 Nov 2019

PONE-D-19-19689R1 

Are primary school children attending full-day school still engaged in sports clubs? 

Dear Dr. Spengler:

I am pleased to inform you that your manuscript has been deemed suitable for publication in PLOS ONE. Congratulations! Your manuscript is now with our production department. 

With kind regards,

on behalf of

Dr David Meyre 

Academic Editor

PLOS ONE